# Source-Aware Training Enables Knowledge Attribution in Language Models

**Muhammad Khalifa**[†*]**, David Wadden**[¶]**, Emma Strubell**[¶§]**,**

**Honglak Lee**[†]**, Lu Wang**[†]**, Iz Beltagy**[¶]**, Hao Peng**[‖*]

University of Michigan[†], Allen Institute for AI[¶], Carnegie Mellon University[§],
University of Illinois Urbana-Champaign[‖]
`khalifam@umich.edu`

## Abstract

Large language models (LLMs) learn a vast amount of knowledge during pretraining, but they are often oblivious to the source(s) of such knowledge. We investigate the problem of *intrinsic source citation*, where LLMs are required to cite the pretraining source supporting a generated response. Intrinsic source citation can enhance LLM transparency, interpretability, and verifiability. To give LLMs such ability, we explore *source-aware training*—a recipe that involves **(i)** training the LLM to associate unique source document identifiers with the knowledge in each document, followed by **(ii)** an instruction-tuning stage to teach the LLM to cite a supporting pretraining source when prompted. Source-aware training borrows from existing pretraining/fine-tuning frameworks and requires minimal changes to the model architecture or implementation. Through experiments on synthetic data, we demonstrate that our training recipe can enable faithful attribution to the pretraining data without a substantial impact on the model's perplexity compared to standard pretraining. Our findings also highlight the importance of pretraining data augmentation in achieving attribution.[1]

## 1   Introduction

Large language models (LLMs) often generate content that is not based on factual information (Ji et al., 2023; Ye et al., 2023a). As LLMs are pretrained over noisy web data that often contains inaccurate or outdated content, users should be able to verify LLM outputs by checking their sources. Moreover, concerns about copyright infringement (Min et al., 2023; Longpre et al., 2023), privacy violations (Kim et al., 2024), data contamination (Shi et al., 2023), and toxic content (Gehman et al., 2020) in LLMs emphasize the need for techniques to identify and trace the origins of information included in models' responses. It is therefore desirable if LLMs can provide supporting evidence

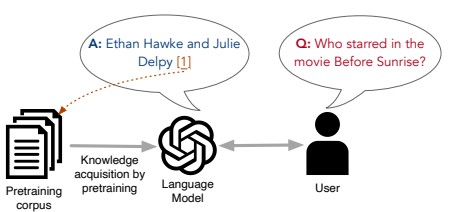

Figure 1: **Intrinsic source citation:** The language model cites pretraining document(s) from which it acquired its relevant parametric knowledge.

for their responses by citing or attributing the outputs to the sources they draw upon (Rashkin et al., 2023; Huang & Chang, 2023; Li et al., 2023b). Beyond improving the models' transparency, attribution allows for a deeper understanding of the relationship between training data and model behaviors, thereby offering a pathway to refine the quality of pretraining data.

We focus on *intrinsic source citation*, where the LLM should cite source documents from the pretraining data from which it acquired its relevant parametric knowledge. Compared to

---

[*]Work partially done while these authors were at the Allen Institute for AI.

[1]Code and data available here: `https://github.com/mukhal/intrinsic-source-citation`.

retrieval-based approaches such as RAG (Lewis et al., 2020; Guu et al., 2020) or post-hoc techniques (He et al., 2023; Gao et al., 2023a), intrinsic source citation is inherently tied to the model itself, enabling more faithful attribution to its parametric knowledge, and opening up unique opportunities for improved interpretability (Alvarez Melis & Jaakkola, 2018; Marasovic et al., 2022).

To this end, we explore *source-aware training* to enable an LLM to cite the pretraining source supporting its parametric knowledge. Our motivation is three-fold. First, a significant portion of an LLM's knowledge is acquired during pretraining, therefore citing evidence for this parametric knowledge can greatly enhance the LLM trustworthiness. Second, the standard practice for LLM pretraining neglects attribution, which explains why LLMs currently fail to provide reliable citations (Agrawal et al., 2023; Zuccon et al., 2023). We aim to explore a training procedure that naturally facilitates citation of the pretraining data. Finally, from a scientific perspective, it is intriguing to investigate whether and how current language models can be trained to reference their pretraining data.

We inquire: Given an off-the-shelf LLM, can we train it to attribute its generations to the supporting sources from the pretraining data? Our goal is to cite the pretraining documents themselves (see Figure 1). Our setup mirrors existing frameworks for LLM (continual) pretraining and can be summarized as follows: We take an off-the-shelf LLM, continue to pretrain it on a corpus associating each document with a unique identifier, then fine-tune it to answer questions about the acquired knowledge while providing citations. The citation is achieved by generating an identifier of a document supporting the answer. Continual pretraining is done as in prior work, with the main difference of injecting the document identifiers into the pretraining data—minimal changes in the model's architecture or implementation are needed.

To study the generalization over this task and simulate a realistic fine-tuning setting, we limit our instruction tuning stage to a subset of the pretraining documents (in-domain) and evaluate the model's attribution ability over the remaining (out-of-domain) documents. We run experiments over a synthetic pretraining corpus of fictitious biographies and show that LLMs can achieve reasonable attribution when answering a question about the out-of-domain documents.

Our contributions are summarized as follows:

- To the best of our knowledge, this work is the first to study intrinsic source citation and investigate the ability of current LLMs to cite the source of their parametric knowledge.

- We explore source-aware training, which we apply to off-the-shelf LLMs to give them the ability to attribute their outputs to the pretraining sources. On synthetic data, we show that such training can achieve reasonable attribution without strongly hurting the LLM perplexity compared to standard pretraining.

- We examine the impact of various training strategies on attribution, such as training to cite the gold document (Section 3.3) and data augmentation (Section 5.3), and our findings can inform future efforts to train attribution-capable models at a large scale.

## 2 Related Work

**Language Model Attribution.** Attribution is gaining more attention recently as interpretability and grounding of language models become increasingly important. Generally speaking, approaches to achieve attribution can be classified as either **retrieval-based** or **model-based**. Retrieval-based approaches include retrieval augmentation (RAG) (Lewis et al., 2020; Guu et al., 2020; Borgeaud et al., 2022; Izacard et al., 2023) and post-hoc attribution (He et al., 2023; Gao et al., 2023a). RAG approaches enable attribution by providing a retrieved context for the LM to use, and teaching LM how to cite the retrieved context (Nakano et al., 2021; Menick et al., 2022). The major limitations of RAG approaches are the lack of guarantee that the model is relying on the retrieved data for generation (Petroni et al., 2020; Li et al., 2023a), and that they only work on non-parametric knowledge. Post-hoc approaches (He et al., 2023; Gao et al., 2023a) attribute the LM outputs by retrieving the sup-

porting evidence given the model's response, but have been shown to produce non-accurate citations (Liu et al., 2023).

Model-based techniques involve prompting the model directly to generate citations for its parametric knowledge (Weller et al., 2023; Zuccon et al., 2023) or scaling techniques such as influence functions (Koh & Liang, 2017) to large models (Grosse et al., 2023). Model-based attribution is arguably more faithful than retrieval-based approaches as the citation mechanism is intrinsic to the model (Alvarez Melis & Jaakkola, 2018; Marasovic et al., 2022). However, standard approaches to pretraining LMs do not take into account the need for the language model to cite its pretraining data, which is where our work comes into play. Bohnet et al. (2022) proposed the task of attributed question-answering and evaluated the attribution performance of different systems using the AutoAIS metric (Rashkin et al., 2023; Gao et al., 2023a). In addition, they fine-tuned PaLM (Chowdhery et al., 2023) to generate both an answer and a URL pointing to Wikipedia page supporting the answer in generative retrieval style (Tay et al., 2022; Wang et al., 2022). Although this setup is similar to ours in that we require the LM to generate the document identifier as well, their setup is basically a variation of RAG where the LM acts as the retriever.

**Citation Generation.** There is a large body of work on the task of citation generation in the scientific domain, where the goal is to cite an appropriate article given a particular context (McNee et al., 2002; Nallapati et al., 2008) or to generate text citing one article in relation to another (Xing et al., 2020; Luu et al., 2020). A relevant work to ours is Galactica (Taylor et al., 2022), which leverages the underlying citation graph in the pretraining data to learn to predict citations given a context. Notably, Galactica is trained to leverage citations of scientific articles in the pretraining data, while our work explores citation of all the pretraining documents, extending beyond scientific articles. Gao et al. (2023b) introduced a benchmark for the automatic evaluation of LM citations and Ye et al. (2023b) proposed a method to improve language model grounding by fine-tuning the language model on responses that are well supported by their citations. However, their setup is restricted to citation of retrieved rather than parametric knowledge.

**Generative Retrieval.** Our work is somewhat related to generative retrieval, where an auto regressive model is trained to act as a retriever in an information retrieval (IR) system (Wang et al., 2022; Tay et al., 2022). Generative retrieval typically relies on a transformer model to map a given query to a document identifier that is likely to contain an answer to the query. While our task also requires the language model to generate a document identifier, we differ from generative retrieval in at least two ways. First, our goal is to generate an identifier pointing to a document containing the already generated answer rather than a document that is likely to contain the answer. Second, generative retrieval merely learns a mapping from query to document identifiers, while our setup is concerned with both acquiring knowledge via the next-word prediction objective over the documents and associating acquired knowledge with its source.

## 3 Source-Aware Training

Our training framework is designed to easily integrate with existing pretraining pipelines. We minimize its deviations from established pretraining practice, and it involves almost no modifications to the model architecture or implementation. Each document in the pretraining corpus is assigned a unique document identifier (ID) and our goal is to train a language model that can respond to user prompts by providing both a response and an ID referring to the source document of the model's knowledge.

**Setup.** Our evaluation follows the attributed question answering setup (Bohnet et al., 2022), where given an input prompt $z$, the LLM output will consist of a tuple $\langle r, c \rangle$ where $r$ is the response (e.g., the answer to a question) and $c$ is the identifier of the document in the pretraining data that supports the answer. Following standard LLM training setups, our recipe has two stages: Pretraining (Section 3.1) and instruction tuning (Section 3.2). Instruction tuning trains the model to be able to attribute the generated responses to

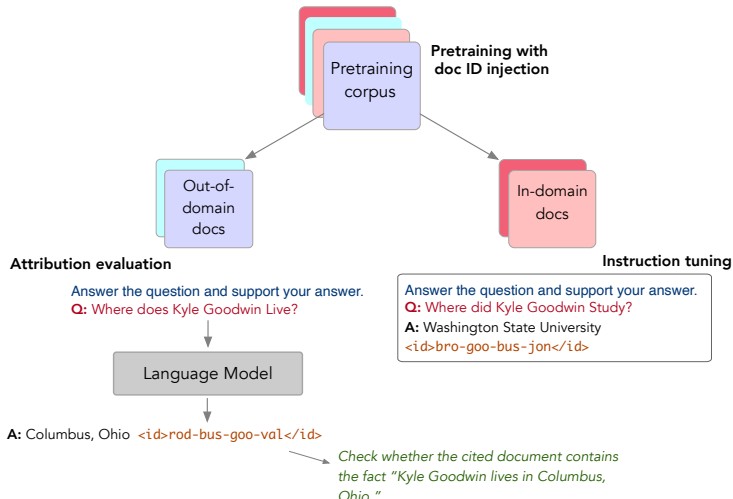

Figure 2: **Training and evaluation setup:** The pretraining corpus is split into in-domain and out-of-domain documents. The in-domain documents are used to create instruction tuning examples, and the out-of-domain documents are used for attribution evaluation.

supporting documents it has seen during pretraining. The pretraining stage will involve all documents by nature, but the instruction tuning step is restricted to a *subset* of the pretraining documents. This restriction is due to the potential cost of curating instruction tuning data from all the pretraining documents, that is in addition to the training overhead incurred by instruction tuning (Zhou et al., 2024).

After training, we measure **out-of-domain (OOD) attribution**: whether the model can attribute knowledge to documents that are only included in the pretraining data but *not* in the instruction tuning data. We therefore split the pretraining corpus into in-domain and OOD subsets. The in-domain data is used to create instruction training examples, while the OOD documents are used for evaluation, as shown in Figure 2.

### 3.1 Pretraining with Doc ID Injection

**Doc ID injection.** The pretraining phase has two goals: **(i)** memorizing knowledge via next-word prediction (same as established LLM pretraining), and **(ii)** associating knowledge within a source document with its ID to enable OOD attribution. We aim to achieve the second goal by *injecting* the document ID into the document before training. An important consideration is the location and frequency of injecting the document ID.

Formally, given a pretraining corpus of documents $\{x^{(1)}, x^{(2)}, ..., x^{(m)}\}$ and their corresponding IDs $\{c^{(1)}, c^{(2)}, ..., c^{(m)}\}$ where each $x^{(i)}$ is a sequence of tokens $x^{(i)} = (x_1^{(i)}, x_2^{(i)}, ..., x_{|x_i|}^{(i)})$, and each $c^{(i)} = (c_1^{(i)}, c_2^{(i)}, ..., c_{|c_i|}^{(i)})$ is a sequence of tokens of its identifier. Our pretraining aims to learn the language model parameters $\theta$ that maximize the objective $\max_\theta \sum_{i=1}^m \log P(\hat{x}^{(i)}; \theta)$, where $\hat{x}^{(i)}$ is the ID-injected version of the document $x^{(i)}$. We inject the doc ID into document a $x$ with different strategies, each of which corresponds to a different $\hat{x}$.[2] Particularly, we experiment with the following strategies:

1. **NO-ID**: Standard pretraining without ID injection: $\hat{x} := x$.
2. **DOC-BEGIN**: Inject the ID once before the first token in the document: $\hat{x} := \{c_1, c_2, ..., c_{|c_i|}, x_1, x_2, ..., x_{|x_i|}\}$.

---

[2]We omit the superscript for brevity.

Answer the question and support your answer.
**A:** Where did Joslyn Valdez study?
**B:** Harvard University ## Ellie Kemp began her career at Apple before joining Oracle in Cary, North Carolina. <id>bus-kem-val</id>

**Chain-of-thought attribution:** generate the answer followed by the rest of the document then the doc ID

Hayden Bush taught at various schools before joining Fox Corporation as a content developer. Joslyn Valdez received her undergraduate degree in Forensic Science from Harvard University. Ellie Kemp began her career at Apple before joining Oracle in Cary, North Carolina. <id>bus-kem-val</id>

Pretraining document with ID injected

Figure 3: Example of chain-of-thought attribution. During both training and inference, the model cites the remaining part of the document before generating the doc ID.

3. **DOC-END**: Inject once after the last token in the document. This is equivalent to $\hat{x} := \{x_1, x_2, ..., x_{|x_i|}, c_1, c_2, ..., c_{|c_i|}\}$.[3]

4. **REPEAT**: Inject the ID after *every* sentence in both in-domain and OOD documents. Here, $\hat{x} := \{X_{s_1}, c_1, ..., c_{|c_i|}, X_{s_2}, c_1, c_2, ..., c_{|c_i|}, ..., X_{s_k}, c_1, c_2, ..., c_{|c_i|}\}$, where $X_{s_j}$ are the tokens in $x$ corresponding to the $j$-th sentence in document $x$ and assuming $x$ has $k$ sentences.

**Disabling cross-document attention.** To maximize GPU utilization during continual pretraining, the typical practice packs several pretraining documents within a single training sequence separated by the end-of-sentence <eos> token. As a result, if cross-document attention is enabled with causal masking, the doc ID tokens for a certain document will attend to preceding tokens from other documents and vice versa. Our initial experiments showed that this severely hurts attribution, since the model will associate the doc ID of a given document with knowledge from other documents in the same training sequence. To avoid this, we disable cross-document attention in all our experiments.

### 3.2 Instruction Tuning

In addition to pretraining, we further adapt the model to **(i)** recall the appropriate knowledge as a response to the prompt and **(ii)** cite the ID of the document supporting the response.[4] This stage does not teach the model any new knowledge, but merely aims at *eliciting memorization* of both the knowledge and doc ID by instruction tuning (Wei et al., 2021; Zhang et al., 2023). Given $l$ examples, the $i$-th example is a tuple $\langle z^{(i)}, r^{(i)}, c^{(i)} \rangle$, where $z^{(i)}$ is the prompt (instruction + query), $r^{(i)}$ is a ground-truth response, and $c^{(i)}$ is the ID of a document that supports the response. The model is trained with the objective $\max_\theta \sum_{i=1}^{l} \log P(r^{(i)}|z^{(i)}; \theta) P(c^{(i)}|r^{(i)}, z^{(i)}; \theta)$. The instruction-tuning examples only come from the in-domain documents, and we use the simple instruction *"Answer the following question and provide evidence for your answer."* Figure 2 shows a fine-tuning example from BIOCITE. During the standard LLM pretraining, i.e., with NO-ID, we remove the doc ID part from instruction tuning examples. Following Taylor et al. (2022), we surround document IDs with two learned special tokens <id> and </id> during both pretraining and fine-tuning.

### 3.3 Chain-of-Thought Attribution

The setup in the previous section trains the model to recall the doc ID immediately after the answer. Another setup we explore is where the model is asked to cite the ground-truth document (or part of it) before generating the doc ID. This can be regarded as an instance of chain-of-thought (CoT) (Nye et al., 2021; Wei et al., 2022). In CoT, we inject the document using DOC-END, then the model is trained to cite the rest of the document after the answer up till the doc ID. Figure 3 shows an example of the CoT setup.

---

[3]DOC-END results in the same training objective as in DSI (Tay et al., 2022), where the model is trained to generate the ID given the full document. While this objective was shown to work for the information retrieval setup, we find that it fails to generalize in attribution.

[4]The instruction tuning examples are curated from the pretraining data such that for a given prompt, we already have the reference document the model should cite. More details are in Section 4.

---

**Document:** Marleigh Austin works at SpaceX. Marleigh Austin studied at the University of Arkansas, Fayetteville. Isaiah Brown studied Graphic Design. Isaiah Brown was born on October 19, 1930. Lora Johnston was born on May 30, 1989. Lora Johnston works at Microsoft Teams. Kyle Goodwin studied at Washington State University. Kyle Goodwin works at Campari Group.
**Doc ID:** `bro-goo-aus-joh`

---

**Instruction tuning example:**
**Q:** Where does Lora Johnston work?
**A:** Microsoft Teams ## `<id>bro-goo-aus-joh</id>`

---

Table 1: An example document in BIOCITE with its unique identifier and an example question. Documents in BIOCITE are constructed by sampling biographies of fake individuals (highlighted in color) from BioS (Zhu & Li, 2023) and then sampling several facts from each biography.

## 4  Data

To have a controlled experimental setting, we rely on pretraining knowledge in the form of *atomic synthetic facts*. We now describe how we construct BIOCITE—a synthetic pretraining corpus.

**Sampling facts.**   BIOCITE is based on the BioS dataset (Zhu & Li, 2023), a collection of biographies of fake people where each biography lists six different facts about each person: birthdate, birth city, study major, university, employer, and work city.[5] Each attribute is described using a corresponding template. For example, the birth city fact is described by "`<person name>` was born in `<birth city>`." To avoid co-reference issues when sampling facts, the person's full name is mentioned in all the facts.

To simulate realistic pretraining data that often include facts about different entities, we construct each document in BIOCITE as a collection of facts from at least *two* different biographies in BioS. More specifically, to construct one document $d$, we first sample the number of biographies $n_d \sim \mathrm{Uniform}([2, \cdots , N_{\mathrm{MaxNBio}}])$. Then, we sample $n_d$ biographies from BioS without replacement. Finally, we sample a random number of facts from each one in the $n_d$ biographies and combine these to form the document. We allow the same combination of biographies to create a document only once and allow each fact to appear only once in BIOCITE.[6]  In our experiments, we generate 100K documents in total using $N_{\mathrm{MaxNBio}} = 4$.

**Creating questions.**   The input prompts for BIOCITE will take the form of factoid questions about the different facts such as "Where does Lora Jonhston Work?". Question generation is done by mapping each fact in the document to a corresponding question template. For example, a fact about a person's birth city is mapped to the question "Where was `<full name>` born?"

**Doc ID format.**   It has been shown that the document ID design plays a role in generative retrieval performance (Tay et al., 2022; Pradeep et al., 2023; Sun et al., 2024) and we observed the same during our initial experiments. When designing a doc ID, we need to be careful not to make the task too easy, where the model can infer the doc ID from the input question without actually performing attribution, but also not too hard, where no semantic overlap exists between the doc ID and the content of the document. Using the last names of the individuals included in a document fits these criteria for two reasons. First, two facts from the same person will most likely exist in different documents. Second, the same last name can be shared by different biographies, whose individuals differ only in the first name. That means relying on the last name will not be sufficient to trivially predict the correct doc ID.

---

[5]Details about reproducing BioS are in Appendix A.1.

[6]In this work, we assume each fact in BIOCITE is mentioned in exactly one document and leave the extension of this work to multi-doc citation to future work.

We choose to use a dash-separated concatenation of the 3-letter prefixes of the last names from the biographies that make up the document, shuffled randomly. Table 1 shows an example document, its ID, and a question extracted from it. Exact dataset statistics are in Table 7 in the Appendix.

**Data augmentation.** LMs struggle to generalize at knowledge extraction over OOD documents (i.e., document that were not seen during fine-tuning) without a sufficient amount of redundancy where the LM will be exposed to the same fact in different formats/positions (Zhu & Li, 2023; Allen-Zhu & Li, 2023; Berglund et al., 2023). In large-scale pretraining setups, this is achieved by scaling the pretraining data but as we study attribution on a smaller scale, we achieve the same effect of redundancy via data augmentation. We mainly apply doc-level augmentation, by shuffling the sentences in each document $N_{\text{aug}} = 3$ times, where $N_{\text{aug}}$ is the number of augmentation samples. Unless otherwise stated, our experiments will include document-level augmentation of the pretraining data, and we will explore the effect of augmentation on attribution in Section 5.3.

## 5 Experiments and Results

### 5.1 Experimental Details

The pretraining corpus is split 50-50 into in-domain and OOD subsets, respectively. Training is done over 80% of the in-domain question, and we show performance in the remaining 20K. OOD evaluation is performed over 20K questions randomly sampled from the OOD documents. The QA performance is evaluated using the token exact match (EM) with the gold answer. During inference, we prompt the model and let it generate a response first, then append the special token `<id>` and continue decoding until the model generates the `</id>` token. We use constrained beam search Cao et al. (2021); Tay et al. (2022) to force the model to generate doc IDs that appeared in the pretraining data.

We evaluate attribution by measuring whether the cited document supports the question-answer pair. Precisely, we measure the gold document ID recall over cases where the answer is correct, where recall is evaluated using Hits@$k$ with $k \in \{1, 10\}$, which measures whether the gold ID is in the top $k$ beams. To monitor the impact of our attribution training on the model quality, we monitor the perplexity over Wikitext-v2 (Merity et al., 2017) during training, as done in previous work (Radford et al., 2019). The model we use for all experiments is TinyLLama 1.1B (Zhang et al., 2024),[7] which we pretrain for 10 epochs with a learning rate of $8 \times 10^{-5}$ and instruction-tuning for 3 epochs with a learning rate of $1 \times 10^{-5}$. During both pretraining and fine-tuning, we apply a linear decay scheduler and use a batch size of 128, a weight decay of 0.02, and a learning rate warm-up of one epoch.

### 5.2 Results

**Downstream QA Performance.** We start by evaluating the QA performance on OOD quuestions. Figure 4 (left) shows answer match over BIOCITE with different document ID injection strategies. The model can achieve OOD answer match > 80%, showing that the model has well memorized the pretraining knowledge. We also note that DOC-BEGIN achieves much worse QA performance than other strategies, and we hypothesize that DOC-BEGIN conditions the model to expect the ID when citing knowledge, causing a mismatch during inference when the ID is absent.

**OOD attribution depends on ID injection strategy.** The ID injection strategy plays a major role in OOD attribution achieved by source-aware training. As shown in Figure 4 (right), placing the ID only once with DOC-BEGIN or DOC-END performs poorly. We hypothesize that both cases train the model to associate the *full* document—rather than individual facts—with the document ID. Precisely, DOC-END conditions the model on the full document when generating the doc ID, but the evaluation requires the model to predict the ID given

---

[7]`huggingface.co/TinyLlama/TinyLlama-1.1B-intermediate-step-1431k-3T`

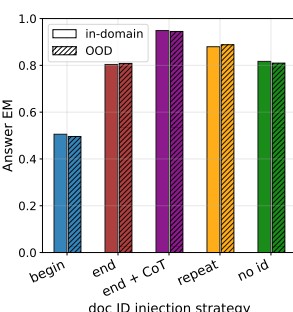 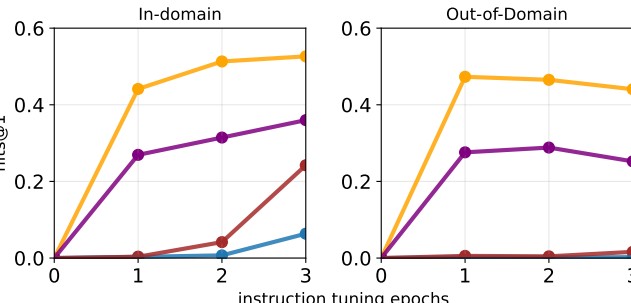

Figure 4: **Left:** Answer EM over questions from in-domain and OOD documents after 1 fine-tuning epoch with different ID injection strategies (Section 3.1). The LLM can generalize well to out-of-domain (OOD) questions in all document ID locations, although both in-domain and OOD answer EM scores degrade with DOC-BEGIN. **Right:** Hits@1 over in-domain and OOD questions during instruction tuning. Only REPEAT and DOC-END + CoT can achieve OOD attribution.

*individual* facts not full documents. This is an instance of LLM generalization failures in knowledge extraction discussed in prior work (Zhu & Li, 2023; Allen-Zhu & Li, 2023) and explains why REPEAT is substantially better, since it trains the model to predict the ID after each fact, making it easier for the model to associate individual facts with the ID.

**Chain-of-thought attribution helps.** REPEAT may be unfavorable, since the number of pretraining tokens will noticeably increase, bringing additional training overhead.[8] Besides, the model quality will be negatively impacted since document IDs are not natural text, which is reflected in the perplexity over Wikitext-v2 shown in Figure 4 (right). The question here is whether source-aware training can yield OOD attribution while injecting the doc ID *once*. Interestingly, the chain-of-thought setup (Section 3.3) achieves reasonable OOD attribution without requiring repeating the doc ID within the document. It is worth noting, however, that the CoT setup adds extra training and inference overhead required to generate chain part of the output. Another interesting observation is that REPEAT and DOC-END + CoT achieve better OOD answer EM compared to NO-ID (e.g., 88.8% with REPEAT vs. 80.9% with NO-ID). We conjecture that source-aware training improves the model grounding to the pretraining data, which reflects on the QA performance.

The results above suggest that source-aware training can teach the model to attribute its parametric knowledge to their pretraining sources, with one key choice to consider: the doc ID injection strategy. Another key component is document augmentation, which we discuss in the next section.

## 5.3 Additional Analysis

**Impact on perplexity.** Now we study the impact of different document ID injection strategies on the LLM quality measured in terms of perplexity over Wikitext-v2. Figure 4 (right) shows perplexity trends during both pretraining and instruction tuning over BIOCITE and Figure 5 (Left) visualizes the tradeoff between LLM quality and OOD attribution.

First, we note that perplexity increases during training in all setups due to the domain shift incurred by training BIOCITE, which does not resemble real text. We can use the perplexity with NO-ID as a baseline and observe how other setups compare to it.

As expected, REPEAT exhibits the worst perplexity, since frequent ID injection means training on more non-natural text. We also note that DOC-BEGIN shows very high perplexity even though the doc ID is injected once, showing that it is best to include the doc ID later rather than earlier in the document. Finally, even though DOC-END + CoT leads to worse perplexity than NO-ID, it is still substantially better compared REPEAT and is Pareto-optimal as shown in Figure 5 (Left). These results that DOC-END + CoT strikes the best balance between OOD attribution and maintaining the model's quality.

---

[8]The number of pretraining tokens increased by about 80% with REPEAT compared to NO-ID.

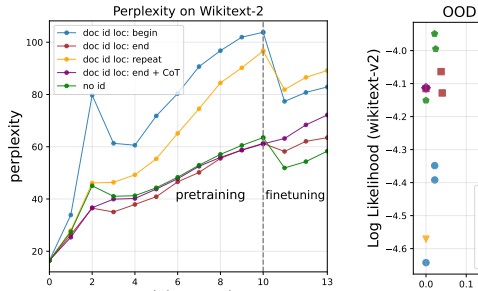 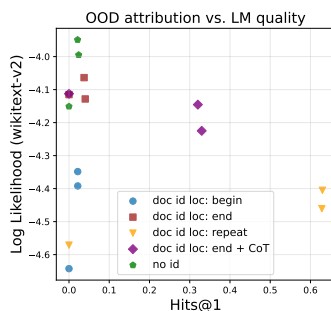 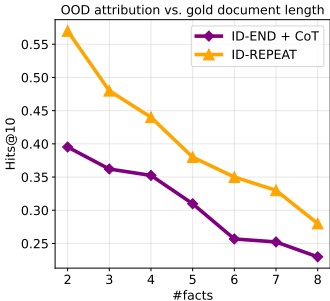

Figure 5: **Left:** LLM quality vs. OOD attribution. Higher is better for Hits@1 and Log Likelihood. Optimal is top-right corner. DOC-END + CoT is Pareto-optimal as it strikes the best balance between LLM quality and OOD attribution. **Right:** OOD attribution performance with different gold document lengths.

**OOD attribution vs. document complexity.** We analyze how OOD attribution varies with the complexity of the document measured in terms of the number of facts when training with REPEAT and DOC-END + CoT. In Figure 5 (Right), we plot OOD attribution measured with Hits@10 changes as the number of facts in the gold document changes. We observe a consistent trend where documents with more facts are harder to cite. This can be explained by the limited representational capacity of the doc IDs: Documents with more facts require the doc ID to be associated with more knowledge.

**Impact of Document Augmentation.** We compare two types of data augmentation methods: document and fact augmentation, and the goal is to assess which type of augmentation is necessary for OOD attribution. Document augmentation is done by permuting the facts within a document $N_{aug}$ times and is what our experiment so far have relied on. Fact augmentation duplicates the facts in a document in $N_{aug}$ different random documents. Figure 6 shows OOD answer match and Hits@1 as $N_{aug}$ is varied and where $N_{aug} = 1$ means no augmentation. While answer match improves using fact-level augmentation, Hits@1 remains the same and only improves when we apply document augmentation. Document augmentation appears necessary for the model to associate the doc ID with the facts in the document.

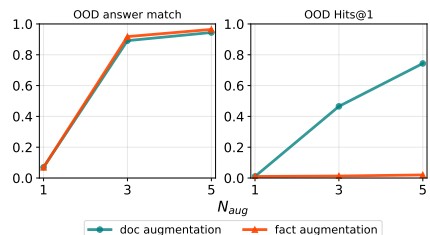

Figure 6: The effect of document- and fact-level augmentation on OOD answer match and Hits@1. $N_{aug} = 1$ means no augmentation applied. We show results when training with the doc ID injection strategy REPEAT.

**Random doc IDs.** To study the effect of the doc ID design, we run experiments with randomly generated doc IDs, where each ID is an 8-digit string (e.g., 53476421). We compare the performance of random IDs to our last name-based design when using REPEAT. Table 2 shows the out-of-domain answer exact match and hits@1 on BIOCITE. While the model still acheives reasonable attribution with random IDs, they underperform our truncated last name design since it is harder for the model to associate numerical tokens (e.g., the token "3") than natural language tokens (e.g., "rog") with document facts.

|  | EM | Hits@1 |
|---|---|---|
| Last names | 88.8 | 47.3 |
| Random ID | 95.9 | 39.8 |

Table 2: Out-of-domain QA and attribution performance with different document ID designs.

**Is the model merely inferring the doc ID from the last name in the input?** Since the doc IDs are constructed as a concatenation of the first three letters of last names in the facts in the documents, the LLM could cheat by predicting doc IDs that contain the prefix of

| **Q:** What university did Adelyn West attend? | **Q:** In what city does Alissa West work? |
| --- | --- |
| **Answer:** University of Pittsburgh. 
 **Top predicted ids:** 
 `jen-lyn-wes` 
 `jen-wes-bur` 
 `jen-cob` | **Answer:** New Orleans. 
 **Top predicted Ids:** 
 `wes-gri` 
 `wes-mcc` 
 `wes-wat-vau` |
| **Gold document:** Adelyn West was born on August 7, 1954. Alissa West lives in New Orleans. Adelyn West studied at University of Pittsburgh... | **Gold document:** Angelina Grimes was born on December 27, 1916. Angelina Grimes studied at... |

Table 3: Two out-of-domain input questions with the same last name and the model predictions. The model predicts totally different doc IDs showing that the model is relying on the fact content itself rather than merely predicting IDs that contain the last name prefix.

the last name in the question. If that were the case, then given two questions about two individuals with the same last name (e.g., Q1: when was Maria Thomson born?; Q2: where does Jake Thomson live?), their generated doc IDs would significantly overlap. We measure this by predicting the top 10 doc IDs for every out-of-domain question. Then for every pair of questions about individuals sharing the same last name (there are 352K such pairs), we compute the overlap between the top 10 predicted doc IDs corresponding to each question using Jaccard index. The average Jaccard over all such pairs is only 0.08, indicating that this is not the case. Table 3 shows two examples of such outputs and the top three predicted doc IDs for each question.

## 6 Limitations

Our work presents a proof-of-concept (PoC) on source-aware training and, as with all PoCs, it has limitations:

- **Synthetic data:** We rely on synthetic rather than real-world data, and the main motivation for this is to control for potential confounding factors introduced by using real data, and which might indirectly affect attribution. Another limitation is that we restrict the form of knowledge to be attributed to factual world knowledge, which we particularly choose since the utility of supporting factual knowledge is more obvious compared to other types of knowledge such as commonsense knowledge, for example.
- **Small-scale experimentation:** Our experiments are done using a relatively small pre-training corpus and model size. This is mainly due to the massive compute that would be required to run hundreds of experiments using a billion-scale pretraining corpus. Nonetheless, we believe the insights revealed by our experiments are valuable and can benefit future research involving large-scale experiments.
- **Cost of source-aware training:** Our experiments show that due to inherent limitations with LLMs, generalization to out-of-domain-documents requires data augmentation, which may practically increase the cost of pretraining. One workaround is to realize that *not all* pretraining data should be cited. For instance, we could select sources that we know to be reliable (e.g., Wikipedia) and only apply source-aware training to these.

## 7 Conslusion

In this work, we studied intrinsic source citation, where models are required to provide support for their parametric knowledge by citing evidence from the pretraining data. We explored a two-stage training process that involved training with document identifiers injected into the pretraining data and then instruction tuning. Our results showed that such source-aware training can enable parametric knowledge attribution in language models when data augmentation is applied. We believe our findings are valuable for future research on training verifiable and trustworthy models.

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

# A   Data details

## A.1   Reproducing BioS

As BioS Zhu & Li (2023) has not been publicly released, we reproduced our own version as follows. Each biography lists six different facts about each person: birthdate, birth city, study major, university, employer and work city. To fill values for each fact, we start by compiling a list of first names, last names, company names, cities, universities, and majors. Examples and counts of the seed entities are shown in Table 4.

Then we construct a single biography by sampling a unique full name (first and last names could repeat) and then to make up each fact, we sample a random suitable entity. For example, for birth and work cities, we sample a random city and for employers, we sample a company name and so on. For birthdates, we generate a random date in the range: January 1st, 1900 – December 31st, 2099. Each fact is constructed via a corresponding template, shown in Table 5. To construct questions for instruction tuning with BIOCITE, each attribute is mapped to a corresponding template question as shown in Table 6.

| Entity | Count | Examples |
|---|---|---|
| First Name | 851 | John, Mary, David, Aria,... |
| Last Name | 557 | Smith, Johnson, Williams,... |
| Company Name | 284 | Apple, Google, Microsoft,... |
| City | 199 | New York, Chicago, Tampa,... |
| University | 178 | University of Alabama, University of Michigan,... |
| Major | 107 | Industrial Design, Fashion Design, Psychology,... |

Table 4: Counts of the compiled entities used to reproduce BioS Zhu & Li (2023).

| Attribute | Fact template |
|---|---|
| Birthdate | `<full name>` was born on `<birthdate>` |
| Work place | `<full name>` works at `<company>` |
| Work city | `<full name>` lives in `<city>` |
| Birth city | `<full name>` was born in `<city>` |
| University | `<full name>` studied at `<university>` |
| Major | `<full name>` studied `<major>` |

Table 5: Templates used to construct different biograohy facts for BioS.

| Attribute | Question template |
|---|---|
| Birthdate | When was `<full name>` born? |
| Work place | Where does `<full name>` work? |
| Work city | Where does `<full name>` live? |
| Birth city | Where was `<full name>` born? |
| University | Where did `<full name>` study? |
| Major | What did `<full name>` study? |

Table 6: Templates used to construct questions from each document in BIOCITE.

## A.2 Dataset Statistics

Table 7 shows statistics of training and instruction tuning data for both BIOCITE and Wikipedia.

|  | Size |
| --- | --- |
| **Pretraining** | |
| #documents | 100K |
| #facts/sents | 408K |
| #tokens | 5.7M |
| avg. sents per doc | 4.1 |
| avg. tokens per doc | 56.9 |
| **Instruction tuning** | |
| #examples | 186K |
| #tokens | 3.1M |

Table 7: Statistics for the BIOCITE before data augmentation. After data augmentation with $N_{\text{aug}} = 3$, the number of pretraining tokens goes up to 17.1M.

