# OpenReview forum: "Source-Aware Training Enables Knowledge Attribution in Language Models"
_colmweb.org/COLM/2024/Conference — COLM_

### Official Review · Reviewer_a1pn · 2024-05-11

**Rating:** 6
**Confidence:** 3
**Ethics Flag:** 1

**Summary:**

This paper introduces a training recipe to elicit large language model to cite from pre-training sources. The proposed method starts with injecting document ids to a set of documents and continue pre-training on the documents with their ids, then perform instruction fine-tuning on set of prompts + document + document ids. The author validate their method in a carefully controlled synthetic setting and found that the proposed method effective improves attribution to pre-training data.

**Reasons To Accept:**

1. The study on pre-training source attribution is very timely — increasing the faithfulness of language model to parametric knowledge can make the output more verifiable and factual, leading to less hallucinations.
2. The proposed method is simple, does not induce too much overhead compared to pre-training (since it just continue pre-trains and do instruction SFT).
3. The experiment results shows the effectiveness of the method compared to the baseline, and the ablations on the location of the id should be of interest of many practitioners.

**Reasons To Reject:**

1. My first concern is that the author only conducted experiments in a **synthetic** setting and the construction of doc id relies on understanding the distribution of the documents, thus whether the method generalizes to real settings is questionable. Moreover, the author only conducted experiments on a small language model, whereas in larger models, it might be easier to make the model produce quotes from pre-training data just with lightweight preference optimization since a larger model memorizes better.

2. The controlled synthetic setting pre-trains a language model on a fixed set of high quality documents, but in reality, the documents are often very noisy. So whether the method generalizes to real language model pre-training is questionable.

Given the above two points, I believe that this paper is novel and studies an important topic, but it is only limited to a very controlled setting.

---

> ### Author Rebuttal · Authors · 2024-05-31
>
> Thank you so much for acknowledging the novelty and importance of our paper! We will address your concerns below:
>
> * **Synthetic setup:** Our motivation for using synthetic data is to control for different confounding variables that may affect attribution, leading us to the wrong conclusions. Our controlled setup allows us to take a step towards tackling this novel task and our and to identify key components that affect attribution such as doc ID injection strategy and data augmentation. Nonetheless, we believe that our approach and findings are valuable to large-scale pretraining setups. We note that **reviewer Szuz** agrees that “it is easy to see how some version of this method could be incorporated into full-scale LLM training.”
> * **Model size:** Our model is 1.1B and while this is considered small by today’s standards, our computational budget did not allow us to experiment with larger models. We agree that scaling the model size will improve memorization and therefore attribution and we leave such exploration to future work.
> * **Noise in real data:** We hope future research could build on our work and scale this to billion-scale pretraining corpora. Nonetheless, our work is a first step in the direction of building more verifiable LLMs with intrinsic attribution capabilities, and we believe our experiments and findings are valuable to the research community.
>
> * **Results on real data**: To alleviate concerns about the applicability of our approach to real data, we have applied source-aware training to 50K docs from [CCNews](https://huggingface.co/datasets/vblagoje/cc_news). For instruction tuning and evaluation, we prompt ChatGPT to generate question-answer pairs from the articles. We used the exact same training hyperparameters as in the paper. Here are the results:
>
> |                           | In-domain |        | Out-of-domain |        |
> |---------------------------|:---------:|:------:|:-------------:|:------:|
> | Doc ID injection strategy | EM| Hits@1 | EM| Hits@1 |
> | ID-BEGIN| 26.2      | 43.7   | 21.0 | 12.5|
> | ID-END| 24.5 | 46.6   | 23.3| 12.1|
> | ID-REPEAT| 27.8 | 64.7   | 27.8| 36.2|
>
> We observe the same performance trend as with BioCite regarding the doc ID injection strategies: ID-REPEAT performs best, and our approach achieves reasonable OOD attribution on a pretraining corpus of real data.
>
> We hope we have addressed your concerns, and we hope you can reconsider your score. Please let us know if you have any more questions!

---

> > ### Comment · Reviewer_a1pn · 2024-06-04
> > **Response by Reviewer**
> >
> > I would like to thank the authors for providing a thoughtful response. I think the authors have addressed my concern on real world data, though I would prefer to see if only adding instruction SFT is able to let a pre-trained LLM output doc ids for a subset in its training data.
> >
> > However, I would like to maintain my score due to the two following concern:
> > - As the model size gets larger and larger, we might not need this kind of effort to solicit the model to output doc ids, since it can memorize more.
> >
> > - I know that not many people have resources to train a 1.1B model, that is why I think more lightweight methods with just SFT or preference optimization on off-the-shelf LLM (e.g. LLama3) would be more impactful. As you have mentioned in your abstract, you introduced a "**post** pre-training recipe", which somewhat indicates that this can be applied to any **pre-trained** LM.
> >
> > Again, I think that the author did well in addressing some my concerns (synthetic v.s. real data), and I think the paper is novel and well written. But I still want to maintain my score due to the computational requirements and scalability of this method. It would make this paper much stronger to see if only adding doc ids to a few trusted sources and applying this method only during SFT to off-the shelf models.

---

> > ### Author Response · Authors · 2024-06-04
> >
> > Thank you for taking the time to respond to our rebuttal.
> >
> > > As the model size gets larger and larger, we might not need this kind of effort to solicit the model to output doc ids, since it can memorize more.
> >
> > Recent advances has shown that memorization even for larger models is challenging. For instance, LLMs such as ChatGPT and GPT-4 still suffer from citation hallucination [1,2]. As we argue in the introduction, this is mainly because their pretraining throws away useful metadata information such as document hyperlinks or URLs, and which our work aims to make use of via principled doc ID injection into the pre-training data.
> >
> > > I know that not many people have resources to train a 1.1B model, that is why I think more lightweight methods with just SFT or preference optimization on off-the-shelf LLM (e.g. LLama3) would be more impactful.
> >
> > Pure SFT or preference optimization could only get you so far if the model has no knowledge of the document IDs it should cite. In other words, SFT/Preference optimization can not train a model to cite a document whose ID it has not seen before, and scaling SFT/preference optimization to all pretraining data will be at least as computationally expensive as our approach.
> >
> > Lastly, our approach is a post training (fine-tuning) recipe that is applied to an already trained TinyLlama model rather than requiring much more expensive training from scratch.
> >
> > We hope our response above has addressed your concerns.
> >
> > **References**
> >
> > [1] Do Language Models Know When They're Hallucinating References? arXiv:2305.18248, 2023
> >
> > [2] Chatgpt hallucinates when attributing answers. Proceedings of the Annual International ACM SIGIR Conference on Research and Development in Information Retrieval in the Asia Pacific Region, 46-51, 2023

---

> > > ### Comment · Reviewer_a1pn · 2024-06-05
> > > **Reviewer Comment**
> > >
> > > Thank you for the response.
> > >
> > > I think that your response have partially resolved some of my concerns. It is true that even ChatGPT / GPT-4 can hallucinate citations. My point was not if we scale up, the model will not hallucinate. My point is that larger models are able to memorize more, and it is easier to elicit larger models to spit out its training data. However, there is one point that I overlooked: even if the models are able to provide verbatim quotes, we still don't know the origin of the sources, which is what the proposed method aim to solve.
> > >
> > > On the second point, I would like to raise that it might not be necessary to produce cites on **all pre-training data**. Producing cites on few trusted sources (e.g. wikipedia) is able to improve the trustfulness of model generations by a significant amount.
> > >
> > > Since there is a point that I overlooked and I believe it is important to produce not just quotes, but also cite sources, I would like to raise my score (5 -> 6).

---

> > ### Author Response · Authors · 2024-06-06
> >
> > We are glad you found our clarification useful, and thanks again for your comments and valuable feedback!

---

### Official Review · Reviewer_6yNB · 2024-05-11

**Rating:** 7
**Confidence:** 3
**Ethics Flag:** 1

**Summary:**

In this work the authors propose the task of intrinsic source citation, namely associating pretraining documents with unique IDs that LLMs can be trained to reproduce when generating answers in a QA setting to attribute the source of the fact from parametric knowledge.

This is a simple, clever, and innovative idea and the authors followed it through very thoroughly, with substantial grounding in existing literature and very thorough empirical results. I think it certainly could set other researchers off on a productive path trying related approaches. The paper is well-written and presents a good starting point for people interested in this area.

I have one major issue, though - I'm very confused as to why the Doc IDs would include the names of the participants at all. You note this doesn't give away *all* the information, but why would we be providing the model with any hints whatsoever as to the correct Doc ID? Why is it not just a random hash of characters, or random sequence of unrelated words, or something? Isn't this making the task dramatically easier?

Relatedly, you say each fact only appears in one document, but in data augmentation are the shuffled documents given the same Doc ID as the original? This is particularly concerning given the finding in Figure 4 that augmentation is strictly required for OOD attribution to work at all.

The authors address this question explicitly with a subheading in section 5, "Is the model merely inferring the doc ID from the last name in the input?" But I don't feel that this paragraph is written sufficiently clearly, how exactly did you calculate this Jaccard Index? I'm not currently convinced by this.

Even taking the findings at face value, I am genuinely questioning the long-term usefulness of the methodological approach presented, because even given the simplified, synthetic data you're working with performance drops off very sharply as documents get longer (Figure 5 right). Nevertheless if empirically valid this is still a useful negative result to present, in which case it's less a reason to reject this paper and more just illustrative of a weakness of the method.

But my overall takeaway is I'm not confident we can trust this empirical evaluation and I really just am confused as to why this problem was introduced into these experiments at all. I can't find a justification for it in the paper, only some reasoning about why it's "not that bad" of a problem. But it just isn't necessary to have this problem, right? Am I missing something? I am willing to substantially revise this review if this can be explained in author response.

**Questions To Authors:**

- In the introduction it's confusing to a reader because you talk about your approach as a post-pretraining recipe but then continually refer to enabling the model to attribute to documents seen in pretraining.
- I would define "parametric knowledge" on the first instance because this term is relatively new, just a simple clause like "knowledge encoded in the model parameters" would help
- In section 2, do we actually know RAG relies only on non-parametric knowledge? This doesn't seem necessarily so to me.
- An example in section 4 gives a name as "Lara Jonhston", probably a typo for "Johnston".
- Figure 3 needs a more clear legend, currently it has to be inferred from the x axis of the left panel.
- Figure 5 appears before Figure 4; also in section 5.3 the textual references to figures are confusing and I think typos / bugs.
- I am colorblind and find the figures marginally hard to read because many distinctions rely only on colors that are relatively close to one another; consider additionally using other features like point shape, line style, etc to make the distinctions more clear.

**Reasons To Accept:**

- Solid core idea
- Thorough empirical tests
- Well-written paper

**Reasons To Reject:**

- Potential serious problem with empirical soundness related to Doc IDs containing the names which are the core entity the questions are about
- Unclear whether the approach presented will actually be at all effective with real-world data

---

> ### Author Rebuttal · Authors · 2024-05-31
>
> Thank you so much for your feedback. We are glad you found the paper idea clever and innovative! We address your concerns below:
>
> * **Doc ID design:** We note that our approach could work with any doc ID format. We mainly use truncated last names design since it performs better and converges faster than e.g. a random ID, while not making the task trivial. Below we compare our doc ID design with random IDs under the ID-REPEAT strategy:
>
> || OOD Answer EM | OOD Hits@1 |
> |--|-|-|
> | Last names truncated | 88.8%| 47.3%  |
> | Random | 95.9% | 39.8%|
>
> We observe that our approach still works reasonably well with random numerical IDs.
>
> * **Data augmentation:** We intend to say that each fact is associated with a single doc ID and will revise our wordings to make that clear. We admit that the requirement of data augmentation is a limitation, but as we suggest, it can be mitigated by selectively augmenting target subsets of the pretraining data that we wish the model to cite.
> * **Jaccard index:** We want to verify the model is not merely predicting doc IDs containing prefixes of the last name in the input question. If that were the case, then given two questions with the same last name, their generated doc IDs would significantly overlap. We measure this by predicting the top 10 doc IDs for every out-of-domain question. Then for every pair of questions (there's 352K such pairs) about individuals sharing the same last name (e.g., Q1: when was Maria Thomson born?; Q2: where does Jake Thomson live?), we compute the overlap (using Jaccard) between the top 10 predicted doc IDs corresponding to each question. The average Jaccard over all such pairs is only 0.08, indicating that this is not an issue. We will elaborate on this in the revision.
> * **Doc length effect:** While attribution is harder as documents become longer since more facts need to be associated with the same doc ID, we speculate that this effect may be mitigated if a larger model is used. We will investigate the effect of model size on attribution in the revision to validate this.
>
> > i really just am confused as to why this problem was introduced
>
> We assume you refer to the doc ID problem addressed above. If not, please let us know.
>
> * **Results on real data:** Kindly refer to our response to **reviewer a1pn**, containing new results of applying our approach to CCNews—a corpus of real data.
>
> We hope we have addressed your concerns, and please let us know if there's anything else you’d like us to clarify!

---

> > ### Author Response · Authors · 2024-06-05
> >
> > Thank you for reviewing our paper and for your feedback. We are looking forward to your response, and please let us know if you have other questions/concerns!

---

> > > ### Author Response · Authors · 2024-06-06
> > >
> > > Dear Reviewer 6yNB,
> > >
> > > The discussion deadline is approaching soon, and we would like to know if our response has addressed your concerns and whether you have additional questions.
> > >
> > > Thank you again for your valuable feedback!

---

> > > > ### Comment · Reviewer_6yNB · 2024-06-07
> > > >
> > > > Apologies for my delay here. I appreciate these clarifications from the authors and feel they have substantially addressed my concerns. I have some remaining hesitations that are largely shared with the other reviewers, but overall appreciate the clarity and effort here and have raised my score.

---

### Official Review · Reviewer_Szuz · 2024-05-11

**Rating:** 6
**Confidence:** 4
**Ethics Flag:** 1

**Summary:**

This work explores the important idea of parametric (i.e. only with model weights) attribution of knowledge in LLMs. The authors propose a “post-pretraining” method for tuning an existing model to 1. Memorize a corpus of information associated with unique doc ids and 2. Use that knowledge in a QA task, with citations to the underlying documents. They specifically apply this to a synthetic dataset of biological sketch information.

The paper is reasonably clear and while the general idea of knowledge attribution is certainly extremely in fashion, this method is original (to my knowledge). I find this paper extremely interesting, well-motivated, timely and a natural idea! However, I have some concerns about the brittleness/effectiveness of this method and the underlying claims that this method could be used for general pretraining corpora, rather than a narrow and carefully groomed synthetic task. I do believe this paper is an important step in the correct direction.

**Questions To Authors:**

The section on attention masking is interesting and very important. However, the chosen method (“such that the ID tokens for a given document only attend to tokens from within that document.”) seems inverted - shouldn’t the mask prevent tokens from other documents from attending to the wrong document ID?

Can you share any comments on randomly generated doc IDs, rather than using the truncated last name method?

**Typos:**

- “still substantially better compared REPEAT and is” - should say compared to repeat
- Section 7 header misspells Conclusion
- “Fig- ure 3 (right) shows perplexity trends during both pretraining and instruction tuning over BIOCITE and Figure 5 (Left) visualizes the tradeoff between LLM quality and OOD attri- bution.” **Should state Figure 5 instead of Figure 3 in the first clause?**

**Reasons To Accept:**

This paper is clearly timely and has potential for great impact. LLMs capable of citing source documents would be extremely helpful at mitigating hallucinations and other LLM misbehavior. As the authors note “a significant portion of an LLM’s knowledge is acquired during pretraining”. This paper also conducts an interesting set of experiments on an admittedly artificial setting. The authors study sensitivity to issues like document ID placement, attention masking for packed batches, and effects of data augmentation. The idea is very natural and it is easy to see how some version of this method could be incorporated into full-scale LLM training.

**Reasons To Reject:**

I find the method and experiments in this paper extremely interesting and the authors make important contributions. However, I have concerns about effectiveness and connections to the claims made about the methods.

Primarily, the abstract states “we explore source-aware-training… to teach the LLM to cite a supporting pretraining source when prompted. Source-aware training can easily be applied to pretrained LLMs off the shelf, and diverges minimally from existing pretraining/fine-tuning frameworks.” I do not believe the results here show that this method can “easily be applied” or that it diverges only minimally. The dataset is very synthetic, relying on what appear to be templatic, short, declarative sentences that uniquely express a single fact. The training instance is Table 1 is clearly an extreme divergence from the far messier presentation of facts in real pretraining data. Language in the intro and abstract suggest that this method will easily work on pretraining corpora, only addressing these concerns in the Limitations section (calling the method a proof-of-concept).

Further, the abstract states “...we demonstrate that our training recipe can enable faithful attribution to the pretraining data without a substantial impact on the model’s quality compared to standard pretraining” - this claim is only referenced by perplexity analysis and I believe the results show the opposite of their claim. It is unclear how well the LLMs will work on any other tasks outside the very narrow biography QA setting. The authors use perplexity numbers to support the claim that their results maintain LLM quality. They measure wikitext-v2 perplexity as a measure of overall LLM quality, citing Radford et al. (GPT-2) as precedent. I understand that perplexities of different models and tokenization are not directly comparable, but we can observe relative trends. Figure 5 shows ppl going from ~19 to ~70, while Radford et al. shows ppl ranging from 29 to 18 at 117M parameters and 1.5B parameters respectively. Thus the increase in perplexity from source aware training seems to be FAR larger than the difference between a 100M model and a 1B parameter model (note the authors use a 1.1B model). Ultimately wikitext ppl seems a very weak measure of LLM quality and this method or the claims made about measurements should be revised.

It also appears that Figure 3 shows several of the methods are substantially worse than the baseline of no document id QA (my apologies if I am misunderstanding the figure).

**In conclusion, I believe this paper is very interesting but would strongly benefit from revising some language in the intro and abstract to more accurately reflect limitations of a synthetic study.**

---

> ### Author Rebuttal · Authors · 2024-05-31
>
> We are grateful for your positive feedback on our paper and recognition of the potential impact of our submission! We aim to address your concerns below:
>
>
> * **Language in the intro:** We will revise our language in the intro and abstract to reflect that our results were done on a synthetic setup.
> * **Perplexity analysis:** We believe that _domain shift_, rather than degraded model quality, is the primary reason for the perplexity increase on WikiText. This is due to training on a specialized corpus such as BioCite. Therefore, in Figure 5, we can see that perplexity still increases even with standard no-id pretraining. We argue that it is unfair to compare our perplexity numbers to Radford et al. as their general-purpose pretraining does not cause such domain shift. We believe that scaling our recipe to large-scale, general-purpose data can mitigate this. We will include this discussion, and revise our language to be more specific to perplexity rather than model quality in general.
> * **QA performance:** Figure 3 (left) shows QA performance with different doc ID injection methods. On the contrary, we see better QA performance with our source-aware training compared to standard pretraining + SFT. We discuss this observation in section 5.2 (under chain-of-thought attribution)
> * **Attention masking**: The mask should prevent document tokens from attending to wrong doc IDs and doc ID tokens from attending to wrong document tokens. For example, assume a sequence of two documents that looks like this ($T_{ij}$ is token $j$ for doc $i$ and $U_{ij}$ is doc ID token $j$ for document $i$): [$T_{11},T_{12},T_{13},U_{11},T_{21},T_{22},T_{23},T_{24},U_{21}$]
>
> The attention mask will prevent $U_{21}$ from attending to $T_{11},..T_{13}$ and $T_{21},...,T_{24}$ from attending to $U_{11}$. We will revise this part to make both scenarios clear.
>
> * **Randomly generated doc IDs**: We initially experimented with randomly generated doc IDs, but found them to underperform our truncated last name design since it is harder for the model to associate numerical tokens (e.g., the token ‘3’) than natural language tokens (e.g., ‘rog’) with document facts. The table below shows the performance of each:
> || OOD Answer EM | OOD Hits@1 |
> |-|-|-|
> | Last-names truncated  | 88.8%| 47.3%|
> | Random ID| 95.9%| 39.8%|
>
> We will include results with random IDs in future revisions.
>
> We hope we have addressed your concerns, and please let us know if you have any more questions.

---

> > ### Comment · Reviewer_Szuz · 2024-06-05
> >
> > Thanks for your detailed response and examples. I also found the new results presented to other reviewers helpful. Note that when I stated "The idea is very natural and it is easy to see how some version of this method could be incorporated into full-scale LLM training" the emphasis should be placed on " **SOME** version of this method " - i.e. a version that works in a less synthetic scenario.
> >
> > You also state "We believe that domain shift, rather than degraded model quality, is the primary reason for the perplexity increase on WikiText." But your paper text states "Now we study the impact of different document ID injection
> > strategies on the LLM quality measured in terms of perplexity over Wikitext-v2." Those two sentences do not seem logically consistent with each other, and the intro also states "we show that such training ... while maintaining a
> > good balance with the LLM quality compared to standard pretraining."
> >
> > The explanations in the body of 5.3 make sense but are not consistent with other claims that wikitext-v2 ppl can be used as a measure of LLM quality. Please ensure this is revised!
> >
> > Thanks to the authors for the new information, I will maintain my positive score.

---

> > > ### Author Response · Authors · 2024-06-06
> > >
> > > Thank you for further comments, and we are glad you found our new results to be helpful. We will make sure to incorporate your feedback about LLM quality claims in future revisions. Thanks again so much!

---

### Official Review · Reviewer_AHVg · 2024-05-14

**Rating:** 4
**Confidence:** 4
**Ethics Flag:** 1

**Summary:**

This paper proposes source-aware training, which trains the LLM to embed source document IDs in the pretraining corpus and instruction fine-tunes the model to learn to cite the supporting document in the generated response. Experiments on some synthetically-created data shows that source-aware training can enable faithful attribution to the pretraining data without a substantial impact on the model’s quality compared to standard pretraining.

**Reasons To Accept:**

Compared with retrieval-augmented generation (RAG), source-aware training allows the LLM to learn the attribution during the training, which avoids the difficulty of post-hoc retrieval process.

**Reasons To Reject:**

- However, the authors only evaluated the source-aware training on a synthetic pretraining corpus. It is hard to say whether the method could work when scaling up to real-world pertaining corpus.
- We believe it is hard to scale up this method because it is costly to collect responses with their corresponding supporting documents. Compared with raw pretraining texts which can be easily collected on a large scale, (response, support document) pairs often do not naturally exist. Collecting such data can be time-consuming and costly. Compared with RAG which only requires a relatively small amount of (query, document) pairs to train the retriever, putting the attribution process at the pre-training stage would greatly increase the demand for such (not-easy-to-collect) data.
- In source-aware training, document identifiers (usually numbers) are embedded into the model parameters. However, those document identifiers are specific to certain corpus. So different models might use different representations of document identifiers (e.g., one with a number and one with a random string). This causes problems when using multiple LLMs and it is less flexible than RAG because corpus-specific identifiers are directly stored in the model parameters.
- This paper lacks a comprehensive comparison with the RAG models. It is not quite clear what's the benefit of source-aware training compared with RAG for generating responses with citations.

---

> ### Author Rebuttal · Authors · 2024-05-31
>
> Thank you for your comments on our paper! We aim to address your comments below:
>
> * **Synthetic data:** We conduct our study on synthetic data because we want to control for different confounding factors that could impact attribution, e.g., complexity, noise in real pretraining data, etc. We believe our findings are valuable for large-scale pretraining efforts and **reviewer Szuz** agrees that “it is easy to see how some version of this method could be incorporated into full-scale LLM training”.
> * **Difficulty collecting SFT data:** In fact, we believe it is fairly easy to collect such data. We first note that the SFT data is only collected over a subset (e.g., 30%) of the pretraining corpus, enabling easy the scaling of our recipe to realistic pretraining setups. Also, this data collection can be automated akin to recent works that employ instruction-tuned LLMs to generate pretraining/SFT data.
> * **Specificity of doc IDS:** In practice, one can associate each model with its own id->document mapping, built from its training data, and then it should be possible to map back to the original document. Do you have a specific use case where this remains an issue?
> * **Comparison with RAG:** The reviewer has made a good point that our approach can often be applied to scenarios where RAG is used. However, we would like to highlight that the goal of our work is fundamentally different from RAG: we aim to explore an intrinsic, model-based recipe for attribution rather than claiming superior performance to RAG, which is external to the model and requires integration with IR systems. Therefore, we do not compare to RAG or post-hoc techniques because the results of such comparison are essential to our main point.
>
> To alleviate concerns about the applicability of our approach to real data, we have applied source-aware training to 50K docs from [CCNews](https://huggingface.co/datasets/vblagoje/cc_news). For instruction tuning, we prompt ChatGPT to generate question-answer pairs. We used the exact same training hyperparameters as in the paper. Here are the results:
>
> | Doc ID strategy | EM| Hits@1 |
> |-|------|--|
> | ID-BEGIN | 21.0 | 12.5|
> | ID-END| 23.3 | 12.1 |
> | ID-REPEAT | 27.8 | 36.2|
>
> We find the same trend as with BioCite: ID-REPEAT performs best and our approach achieves reasonable OOD attribution on a pretraining corpus of real data.
>
>
> We hope we have addressed your concerns, and we hope you can reconsider your score. Please let us know of any more concerns!

---

> > ### Author Response · Authors · 2024-06-01
> > **Typo correction**
> >
> > Dear reviewer,
> >
> > We would like to correct a typo in our rebuttal above in the RAG comparison bulletpoint. Here's the corrected version:
> >
> > * **Comparison with RAG:** The reviewer has made a good point that our approach can often be applied to scenarios where RAG is used. However, we would like to highlight that the goal of our work is fundamentally different from RAG: we aim to explore an intrinsic, model-based recipe for attribution rather than claiming superior performance to RAG, which is external to the model and requires integration with IR systems. Therefore, we do not compare to RAG or post-hoc techniques because the results of such comparison are **not** essential to our main point.

---

> > > ### Author Response · Authors · 2024-06-05
> > >
> > > Thanks again for reviewing our paper. We are looking forward to your response, and please let us know if you have other questions/concerns!

---

> > > > ### Author Response · Authors · 2024-06-06
> > > >
> > > > Dear Reviewer AHVg,
> > > >
> > > > The discussion deadline is approaching soon and we would like to know if our response has addressed your concerns and whether you have additional questions.
> > > >
> > > > Thank you again for your review!

---

### Author Response · Authors · 2024-06-07

We are grateful for all the reviewers' constructive and valuable comments. We are glad the reviewers have found our paper to be well-motivated, timely and with a potential for great impact, and our approach to be original and novel. Based on the reviewers’ feedback, we have made the following main changes:

* **Experiments on real data**: During the rebuttal, we added results on real data in the form of news articles from CCNews. We will expand and include these results in future revisions in addition to our results on synthetic data.

* **Results with different document ID formats**: We have clarified our choice of document IDs and additionally provided results with random IDs, to show that our approach works with different document ID formats.

* **Study model size effect**: Some reviewers asked about how the attribution performance scales with the model size. As we could not provide such results due to the limited time of the rebuttal, we will experiment with different model sizes in the revision.

---

### Decision · Program_Chairs · 2024-07-10

**Decision:**

Accept

**Comment:**

The authors propose a way to perform attributions to pertaining documents via a model's parametric memory - document IDs are included in the pertaining corpus and then fine-tuning is used to extract these at test time.

The major point of contention for all reviewers is the synthetic nature of the experiments, and whether these experiments demonstrate the viability of the approach in a more 'real world' setting. I concur with the general view expressed by Reviewer Szuz -- the paper presents very interesting ideas, ones that will likely be of interest to others and advance the field, though it is unclear if the evidence presented in this paper demonstrates broader applicability.

The authors have committed during rebuttals to be more careful and accurate with claims, highlighting the synthetic nature of their setup and the limited conclusions that can be drawn from such a setup. I believe this is sufficient to address the major weakness of the paper - the lack of immediate real world viability is not, in and of itself, a critical factor to reject a paper if the rest of the paper is of significant interest and the claims are accurately made.

[At least one review was discounted during the decision process due to quality]